# Rheological Properties and Emulsion Stability of Peach Gum Polysaccharides with Different Molecular Weights

**DOI:** 10.3390/foods14193341

**Published:** 2025-09-26

**Authors:** Haoyu Si, Dongmei Zhang, Fan Xie, Songheng Wu, Bingjie Chen, Xiao Wang, Dapeng Sun, Zhan Lin, Yongjin Qiao, Yi Zhang

**Affiliations:** 1Crop Breeding & Cultivation Research Institute, Shanghai Academy of Agricultural Sciences, Shanghai 201403, China; sihaoyu1023@126.com (H.S.); wsh_magnus@163.com (S.W.); chenbingjie0204@126.com (B.C.); wangxiao.0127@163.com (X.W.); sundapeng@saas.sh.cn (D.S.); 2College of Food Sciences & Technology, Shanghai Ocean University, Shanghai 201306, China; 3School of Perfume and Aroma Technology, Shanghai Institute of Technology, Shanghai 201400, China; dmzhang@sit.edu.cn; 4School of Health Science and Engineering, University of Shanghai for Science and Technology, Shanghai 200093, China; xiefan246141@163.com; 5Department of Ecology, Hebei University of Environmental Engineering, Qinhuangdao 066102, China; linzhan1232024@163.com; 6Shanghai Runzhuang Agricultural Technology Co., Ltd., Shanghai 201415, China

**Keywords:** peach gum, thermal extraction, enzyme extraction, rheological properties, emulsion stability

## Abstract

Peach gum polysaccharide (PGP), a natural biopolymer extracted from the resin of the peach tree, holds significant potential for applications in food, cosmetics, and pharmaceutical industries. However, detailed analysis and exploration of its physical and chemical properties remain limited. This study investigates the physicochemical properties, rheological behavior and emulsion stability of PGPs extracted using thermal (TPGP) and enzymatic (EPGP) methods. The results indicate that both polysaccharide fractions exhibit similar arabinogalactan (AG) structures, with high contents of arabinose and xylose, as evidenced by FTIR spectra and monosaccharide composition. However, high-performance size-exclusion chromatography (HPSEC) revealed differences in molecular weights and chain conformations, leading to distinct rheological behaviors. PGP solutions exhibited pseudoplastic flow behavior, with TPGP demonstrating higher viscosity due to its larger molecular weight (1.295 × 10^7^ g mol^−1^). As the PGP concentration increased, gel strength and emulsion stability improved significantly. This study provides more insight into the rheological and emulsifying characteristics of PGPs extracted by varied methods, facilitating their potential applications in food industries.

## 1. Introduction

Peach gum, also called peach resin, is an exudate from peach tree (*Prunus persica*) by mechanical injury or microbial attacks, which is known for its various applications and bioactive properties [1]. Peach gum is rich in polysaccharides, which are the main components responsible for its diverse functional properties. Peach gum polysaccharides (PGPs) have attracted extensive research interest due to their potential in numerous industrial applications, such as the pharmaceutical field, food industry, adsorbent materials, binders, and so on [2,3]. PGP is a complex carbohydrate composed of monosaccharide units such as galactose, rhamnose, glucose, and arabinose [4]. The structural characteristics of these polysaccharides, such as molecular weight and degree of branching, played a pivotal role in determining bioactive properties, including antioxidant, antitumor, and immunomodulatory effects [5].

The structure and conformation of polysaccharides are influenced by variations in molecular weight and extraction methods, which in turn affect their physical properties and biological activities. PGPs extracted by alkaline hydrolysis and hydrogen peroxide were confirmed to have similar main structures. However, the PGPs obtained through alkaline hydrolysis exhibited higher molecular weight, viscosity, and emulsion stability [4]. Additionally, both alkaline hydrolysis and hydrogen peroxide-extracted PGPs demonstrated potent antioxidant and antimicrobial activities [6]. In addition, to obtain PGPs with better solubility, the methods described above were typically used to reduce the molecular weight of crude peach gum. Currently, Yang et al. hydrolyzed peach gum at 140 °C for 2 h, reducing its molecular weight to as low as 6.8 × 10^4^ g mol^−1^ [7]. Notably, excessively high hydrolysis temperatures can lead to further degradation of PGPs into furfural and 5-hydroxymethylfurfural [5]. Enzymatic hydrolysis is characterized by its specificity and mild reaction conditions. However, studies on the enzymatic preparation of PGP are still limited. Evidence from earlier studies indicates that the main polysaccharide chain of PG consists of β-D-galactose units linked by (1 → 6) glycosidic bonds [5]. Therefore, β-galactosidase was selected to produce PGP with lower molecular weights in this study.

PGP solution has been reported as a typical non-Newtonian fluid with good rheological properties and emulsifying capabilities. Factors such as pH, ionic strength, and temperature can significantly affect the rheological properties of PGP [8]. For example, the addition of NaCl to PGP solution alters the interactions between macromolecules in PGP, which leads to aggregation and reduces apparent viscosity [8]. Meanwhile, the emulsifying properties of PGP were influenced by its molecular weight, concentration, and the conformation of macromolecules [9]. According to Wei et al. [10], the emulsion becomes more stable as the concentration of PGP increases, with oil–water emulsions showing greater stability when the concentration surpasses 5%. In addition, PG has better emulsifying ability and stability than gum arabic and has a certain potential in food and cosmetics applications as emulsifiers and stabilizers [4]. However, Wei et al. and Chen et al. investigated the function of PGP under normal conditions, neglecting the potential impact of external processes, including thermal treatment, and the research on the effect of peach gum on emulsion at different storage temperatures is still limited [10,11]. Therefore, investigating the impact of heating on PGP obtained through different extraction methods and the effect of peach gum on emulsion at different storage temperatures are of significant importance for industrial applications.

The molecular weight of tree gum polysaccharides is another critical factor influencing their functional properties. High-molecular-weight polysaccharides generally exhibit higher viscosity and better emulsifying capacity, while low-molecular-weight polysaccharides tend to have better solubility and permeability. By manipulating the molecular weight distribution, it is possible to tailor the properties of PGP for specific applications. This study aimed to investigate the molecular and physical characteristics of PGP, with particular emphasis on the rheological properties and emulsification of PGPs of varying molecular weights with different extraction methods, which aims to provide insights for further research and application of PGP.

## 2. Materials and Methods

### 2.1. Materials

Peach gum exudates were collected from the trunks and branches of peach trees (Prunus persica Batsch.) in the peach orchard of Guoyuan Village, Datuan Town, Nanhui District, Shanghai. The monosaccharide standards (glucose, galactose, mannose, arabinose, rhamnose and xylose) and galacturonic acid were from SigmaAldrich (St. Louis, MO, USA). Soybean oil was purchased from Yihai Kerry Arawana Holdings Co., Ltd. (Shanghai, China). Whey protein isolate (CAS: 84082-51-9) and β-galactosidase (CAS: 9031-11-2) were purchased from Shanghai Yuanye Bio-Technology Co., Ltd. (Shanghai, China). Arabic gum (derived from acacia trees, CAS: 9000-01-5) was obtained from Shanghai Macklin Biochemical Technology Co., Ltd. (Shanghai, China). The other reagents employed in this study were of analytical grade.

### 2.2. Preparation of PGP

#### 2.2.1. Thermal Extraction and Purification

The thermal extraction (Figure 1) of PGP was based on a method described by Wei et al. [10] with modifications. A suspension was prepared by adding 10 g of PG to 1 L of distilled water, soaking the mixture at 60 °C for 24 h, and homogenizing it with a blender. After stirring at 90 °C and 500 rpm for 2 h, the suspension was centrifuged at 4000 rpm for 20 min to collect the supernatant and repeated twice. The supernatants were combined and reduced to 20% of the original volume by concentration, and then absolute ethanol was added at a 4:1 ratio and left to stand for 8 h. The solution was freeze-dried to obtain TPGP.

#### 2.2.2. Enzymatic Extraction and Purification

A suspension was prepared by adding 10 g of PG to 1 L of distilled water, soaking the mixture at 60 °C for 24 h, and homogenizing it with a blender. The pH was adjusted to 7, followed by the addition of β-galactosidase at 176 U/g. The mixture was stirred at 50 °C and 500 rpm for 24 h and centrifuged at 4000 rpm for 20 min to collect the supernatant. After centrifugation, the pellet was discarded, and only the supernatant was further processed for ethanol precipitation. The supernatant was concentrated to 20% of the original volume, and then absolute ethanol was added at a 4:1 ratio (*v*/*v*) and left to stand for 8 h. The solution was freeze-dried to obtain EPGP (Figure 1). And The extraction yield of PGPs was determined using the below equation:(1)Yield (%) = *W*_1_/*W* × 100%, where *W*_1_ indicates the dry weight of the extracted PGPs (g) and *W* indicates the dry weight of peach gum (g).

### 2.3. Physicochemical Characteristics

#### 2.3.1. Chemical Component

The total sugar content of TPGP and EPGP was determined by the phenol–sulfuric acid method, while the uronic acid content was measured using the m-hydroxydiphenyl method [10].

The determination of monosaccharide composition was based on a method described by Safdar et al. with some modifications [12]. PGP was dissolved in 2 mol/L trifluoroacetic acid (TFA) at the ratio of 1 mg: 1 mL and hydrolyzed at 121 °C for 2 h. Following complete hydrolysis, 200 µL of methanol was added in three steps under nitrogen protection at 50 °C to ensure the complete removal of TFA. The hydrolysate was diluted to a final volume of 10 mL with distilled water, and the aqueous phase was collected and filtered through a 0.22 µm aqueous-phase membrane. Monosaccharide composition was determined using a Thermo ICS 5000+ ion chromatography system (Thermo Fisher Scientific, Waltham, MA, USA) equipped with an electrochemical detector.

The chromatographic analysis employed a Dionex™ CarboPac™ PA20 column (150 × 3.0 mm, 10 µm) with a 5 µL injection volume. The mobile phases were solvent A (H_2_O), solvent B (0.1 M NaOH), and solvent C (0.1 M NaOH, 0.2 M NaAc), delivered at a flow rate of 0.5 mL/min, with the column temperature maintained at 30 °C. Different concentration of mixed standard substance (0.4–40 μg/mL) were prepared to establish a standard curve and analyze the monosaccharide compositions.

#### 2.3.2. Molecular Weight Analysis

The molecular weight parameter was measured according to the method previously described by Zhang et al., with some modifications [13]. Dissolve 5 mg of PGPs completely in 0.1 M NaNO_3_ aqueous solution (containing 0.02% NaN_3_) at a final concentration of 1 mg/mL and filter through a 0.45 μm membrane filter. The mobile phase consisted of an aqueous solution containing 0.1 M NaNO_3_, 0.02% NaN_3_ at a 0.6 mL/min flow rate. Molecular weight distributions of the samples were determined using the chromatographic system with a differential refraction index detector and a multiangle light scattering detector (DAWN HELEOS II, Wyatt Technology, Goleta, CA, USA). The gel exclusion chromatographic columns Ohpak SB-805 HQ (300 mm × 8 mm) and Ohpak SB-803 HQ (300 mm × 8 mm) were used in series at 45 °C. The value of dn/dc was determined as 0.14 mL/g. Data were acquired and processed using ASTRA6.1 (Wyatt Technology).

#### 2.3.3. Thermogravimetric Analysis (TGA)

The thermal stability of PGPs was determined by a TGA2 thermogravimetric analyzer (Mettler Toledo, Columbus, OH, USA). PGPs were heated from 25 to 600 °C at a heating rate of 10 °C/min, and the flow rate of N_2_ was 20 mL/min. The effect of temperature on sample weight was assessed using roughly 3 mg of material, while an empty aluminum pan was employed as a control.

#### 2.3.4. FT-IR

PGPs were characterized by FT-IR spectroscopy using a Nicolet iS5 instrument (Thermo Fisher Scientific Inc., USA), with spectra recorded from 500 to 4000 cm^−1^ at 4 cm^−1^ resolution.

### 2.4. Rheological Analysis

A certain amount of PGPs powder was weighed and mixed with deionized water to prepare PGP solutions at concentrations of 2, 4, and 6% (*w*/*w*).

#### 2.4.1. Steady Shear Tests

PGPs were tested by a DHR20 advanced rheometer (TA Instruments, New Castle, DE, USA) with a 40 mm cone plate, 0.05 mm gap, and shear rate range of 0.1–1000 s^−1^. The steady flow behavior of TPGP and EPGP solutions at concentrations of 2, 4, and 6% (*w*/*w*) was evaluated at 25 °C.

To investigate the changes in apparent viscosity under different temperatures, TPGP and EPGP solutions at concentrations of 2, 4, and 6% (*w*/*w*) were heated from 25 °C to 85 °C at a rate of 5 °C/min. Measurements were taken at a shear rate of 10 s^−1^ to determine apparent viscosity.

#### 2.4.2. Creep–Recovery Tests

During the creep test, TPGP and EPGP solutions with concentrations of 2, 4, and 6% (*w*/*w*) were subjected to constant stress of 0.02 Pa at 25 °C for 250 s. The stress was abruptly removed in the recovery test, and the samples were held at 25 °C for 250 s. Shear strain variations were monitored continuously throughout both phases of the experiment.

#### 2.4.3. Dynamic Oscillatory Tests

The linear viscoelastic region (LVR) was determined by performing a strain sweep from 1 to 100% at 1 Hz and 25 °C. Subsequently, an oscillatory frequency sweep test was conducted within a frequency range of 0.1–10 Hz at a strain of 10% and a temperature of 25 °C.

#### 2.4.4. PGP Gelling Network Analysis

TPGP and EPGP solutions at concentrations of 2, 4, and 6% (*w*/*w*) were freeze-dried and applied to the conductive adhesive. After gold sputtering, the particle morphology was observed and recorded at magnifications of 300× under an accelerating voltage of 2.0 kV by TM 4000+ SEM (Hitachi, Tokyo, Japan).

### 2.5. Effects of Peach Gums on the Emulsion Stability of Whey Protein Isolate

#### 2.5.1. Emulsion Preparation

Emulsions were prepared based on the method described by Qian et al. [4], with appropriate modifications. A solution was prepared with 2% (*w*/*w*) whey protein isolate (WPI) and 6% (*w*/*w*) PGPs and gum arabic (GA), and thoroughly stirred to ensure complete dissolution. The aqueous phase consisted of deionized water at pH 7, without added buffer. The aqueous and oil phases were preheated to 50 °C prior to homogenization. All gum solutions were combined with 50% (*w*/*w*) soybean oil and homogenized at 16,000 rpm for 3 min by T18 digital ULTRA-TURRAX (IKA, Staufen, Germany) with a rotor of approximately 10 mm diameter. The final composition of the emulsions consisted of 3% PGPs or GA, 1% WPI, and 25% soybean oil. 5% pentanediol was added as an antimicrobial agent. The emulsion temperature after homogenization was maintained at 50 °C.

#### 2.5.2. Centrifuge Stability and Storage Stability

Centrifugation and storage stability were carried out following the method of Naji-Tabasi et al. [14]. The emulsion was centrifuged immediately after preparation at 2000× *g* for 10 min (CR21N, Hitachi, Japan), and emulsion stability (ES) was calculated as:(2)ES (%) = *Ve*/*Vt* × 100%, where *V**e* represents the cream layer, and *Vt* represents the total emulsion height. Two replicates of each sample were prepared.

For the measurement of emulsion storage stability, each sample was transferred to a container and stored at 4 °C, 25 °C, and 45 °C for 5 weeks. Data were recorded, and ES was calculated.

### 2.6. Statistical Analysis

The data were obtained by TRIOS software, and fitting was conducted in Origin 2018. Values are reported as mean ± SD and compared by ANOVA in SPSS 27.0, with significance defined at *p* < 0.05.

## 3. Results

### 3.1. Physicochemical Characteristics

#### 3.1.1. Chemical and Monosaccharides Composition Analysis, Molecular Weight and Rz of PGPs

The yield of TPGP from peach gum (as shown in Table 1) was 52.3% and the yield of EPGP was 46.8%. The total sugar content of TPGP and EPGP were 70.62% and 72.13% tested by the phenol–sulfuric acid method, respectively [15]. The m-hydroxybiphenyl method revealed that TPGP and EPGP contained 4.34% and 2.13% uronic acid, respectively, and low uronic acid content may result in lower solubility [11]. The monosaccharide content in various PGP fractions is summarized in Table 1. The primary components of PGP are Ara and Gal, indicating that PGP belongs to arabinogalactan (AG). The combined content of Ara and Xyl is relatively high, suggesting a highly branched structure for all PGP fractions [16]. The low content of uronic acids in PGP might contribute to its poor water solubility. These findings are consistent with previous studies, although differences in the monosaccharide proportions were observed [4,17].

The molecular weight and conformational parameters of PG extracted by thermal and enzymatic methods are presented in Table 2, with both showing a unimodal distribution. The weight-average molecular weights (Mws) of TPGP and EPGP were 1.295 × 10^7^ and 5.718 × 10^6^ g mol^−1^, respectively. The polydispersity index (*M*_w_/number-average molecular weight (*M*_n_)) of TPGP (1.994) and EPGP (2.070), which reflects the breadth of molecular weight distribution [18], indicates that molecular weight distribution of TPGP and EPGP were moderately polydisperse (Figure 2A). Molecular weight is one of the key factors directly affecting solubility. The molecular weight of EPGP was reduced relative to that of TPGP, indicating that enzymatic hydrolysis can more effectively yield lower molecular weight PGPs compared to thermal extraction.

The z-average radius of gyration (*R*z) represents the spatial extension of polymer chains and reflects the molecular size [19]. *R*z of TPGP was larger relative to that of EPGP, which is consistent with the findings of Hundschell et al. [20], where an increase in molecular weight led to a decrease in polymer density, thereby somewhat affecting its spatial extensibility. It was reported by Lin et al. [21] that some polysaccharides with high molecular weight exhibit complex spatial structures and weaker bioactivity. Therefore, the molecular weight and the hydrodynamic radius of polysaccharides may also influence their biological activity.

#### 3.1.2. FT-IR Spectroscopy

The FT-IR spectra of TPGP and EPGP fractions are illustrated in Figure 2B. TPGP and EPGP exhibited similar spectra profiles. The broad bands at 3258 cm^−1^ and 2930 cm^−1^ were, respectively, attributed to the stretching vibrations of -OH and C-H. The characteristic symmetric and asymmetric stretching vibrations of COO- were observed at 1620 cm^−1^ and 1417 cm^−1^ [22]. The shoulder peak at 1070 cm^−1^ and peak at 1020 cm−1 correspond to β-galactan and AG, respectively, indicating AG’s presence [23]. These results are consistent with those reported by Chen et al. [11] and corresponds to the results of the monosaccharide composition analysis.

#### 3.1.3. Thermal Stability of PGPs

Due to its accuracy and simplicity, thermogravimetric analysis (TGA) is commonly employed in polysaccharide research to analyze the decomposition patterns of polymers [24]. Figure 3 shows the results of TGA and derivative thermogravimetric analysis (DTGA) for the thermal stability of TPGP and EPGP, which reveals three distinct weight loss stages as the temperature increases. The first weight loss (7.5%) occurs between 25 and 100 °C, primarily as a result of water evaporation. The second weight loss of TPGP and EPGP, observed between 235 and 350 °C, is attributed to the molecular degradation of the polysaccharides. At this stage, two distinct peaks were observed for both TPGP and EPGP. The DTGA peak temperatures of TPGP were 252 °C and 325 °C, with corresponding weight losses of approximately 15% and 38%, respectively. For EPGP, the DTGA peak temperatures were 256 °C and 323 °C, with corresponding weight losses of about 10% and 28%, respectively. This suggests that the PGPs contain two major components with different molecular weights or structures. As previously reported by Guo et al., these two components correspond to the acidic and neutral fractions [25].

### 3.2. Steady Flow Behaviors

Figure 4(A-1) showed the effects of different concentrations (2%, 4%, and 6% (*w*/*w*)) on the rheological properties of TPGP and EPGP solutions. The viscosity of all solutions decreased with increasing shear rate. Shear thinning is observed due to the alignment of randomly oriented molecules in the flow direction, which diminishes interactions between adjacent chains [17]. All solutions of TPGP and EPGP with different concentrations exhibited shear-thinning behavior and characteristics of pseudoplastic fluids [26]. As the concentration increased, the growing number of PGPs molecules and the shortened intermolecular distance facilitated collisions, coverage, and overlapping among molecular chains, ultimately resulting in higher apparent viscosity [27]. Therefore, the higher the concentration, the greater the viscosity of the TPGP and EPGP solutions. It demonstrated that the shear rate-dependent flow behavior is governed by molecular weight and factors including polymer conformation [28]. The alignment of molecular chains contributed to the shear-thinning behavior of peach gum solutions. Flexible-chain polymers showed a significant decrease in viscosity with increasing shear rate compared to rigid-chain polymers. As flexible-chain polymers, shear-thinning behavior was observed for TPGP and EPGP, with apparent viscosity rising as molecular weight increased. Due to its higher molecular weight relative to EPGP, TPGP exhibits higher viscosity.

To further investigate the effect of temperature changes during processing on the rheological properties of TPGP and EPGP solutions, Figure 4(A-2) shows the impact of temperature (25 °C to 85 °C) on the rheological behaviors of TPGP and EPGP solutions. At the same concentration, TPGP showed higher viscosity than EPGP, consistent with the well-established correlation between polysaccharide molecular weight and viscosity. As temperature increased, the viscosity of both TPGP and EPGP solutions decreased, likely due to increased flexibility and flowability of the molecular chains [29]. All solutions exhibited shear-thinning behavior, which became more pronounced at higher concentrations. These observations align with previous reports showing that AGs with (1 → 6)-galactan generally exhibit higher viscosity and more pronounced shear-thinning behavior than AGs with (1 → 3)-galactan [30], and that PGP solutions have higher viscosity than GA solutions at comparable concentrations [5].

### 3.3. Creep and Recovery Behavior

Creep–recovery curves of TPGP and EPGP solutions at various concentrations are presented in Figure 4(B-1,B-2). During the creep phase (0–250 s), a constant stress was applied to the solution, and the increase in creep compliance was associated with the deformation induced under stress. However, in the recovery phase (250–500 s), when the stress was instantaneously removed, the strain rapidly decreased and gradually approached a constant value over time.

The creep compliance decreased with increasing concentration, indicating that a stronger intermolecular network was formed as the concentration increased. During the recovery phase, the creep-recovery curves of TPGP and EPGP solutions exhibited typical viscoelastic characteristics, with a reduction in compliance [31]. The results further confirmed that the elastic properties of PGP solutions increase with concentration, which is achieved through enhanced intermolecular networks. Notably, the recoverability of TPGP and EPGP solutions decreased with increasing concentration, which aligns with the results of Guo et al. for flaxseed gums solutions [25]. However, the specific mechanism remains unclear and requires further investigation.

### 3.4. Dynamic Rheological Properties

The solution behavior of polymers is analyzed using dynamic rheology, and the LVR of the polymer gel is determined by strain sweep tests [25]. As presented in Figure 4(C-1), 4%, and 6% TPGP and EPGP solutions predominantly exhibited elastic behavior in the LVR, with G′ exceeding G′′. In contrast, 2% TPGP and EPGP solutions displayed primarily viscous behavior in the LVR. At the same concentration, TPGP solutions with higher molecular weight exhibited higher G′ values, and G′ increased with the concentration of PGPs, which confirmed that higher concentrations facilitated the formation of a more solid-like structure [32]. Furthermore, as the strain increased, the reduction in G′ indicated the disruption of the gelling network [33].

Figure 4(C-2) illustrates the variations in G′ and G′′ with frequency for TPGP and EPGP solutions at different concentrations, further confirming the weak gel behavior of PGP solutions. The G′/G′′ ratio also increased as the concentration rose. The G′/G′′ ratio for 6% (*w*/*w*) TPGP solution ranges from 2.61 to 3.11, higher than that of the 2.0% (*w*/*w*) TPGP solution (2.14 to 2.51) and 4% (*w*/*w*) TPGP solution (2.08 to 2.55). The G′/G′′ ratio for the 6% (*w*/*w*) EPGP solution ranges from 2.37 to 2.75, higher than that of the 2.0% (*w*/*w*) EPGP solution (1.06 to 1.76) and 4% (*w*/*w*) EPGP solution (1.70 to 2.11), which once again suggested that higher concentrations of PGPs promoted the formation of stronger gels [34].

### 3.5. Gel Microstructure

The three-dimensional networks of TPGP and EPGP gels are depicted in Figure 5. The intermolecular networks formed in the TPGP and EPGP solutions are partially preserved after freeze-drying. As the concentration increases, both the density of the gel network and the integrity of the network cells improve. And at the same concentration, TPGP has a higher density of gel networks and a higher integrity of network cells. In conclusion, higher concentrations and molecular weight of PGPs lead to stronger molecular networks. Chen et al. obtained PGP with a molecular weight of 1.31 × 10^7^ g mol^−1^ by performing thermal hydrolysis on crude peach gum at 95 °C for 3 h [11]. Similarly, Wei et al. produced PGP with a molecular weight of 1.64 × 10^7^ g mol^−1^ by thermal hydrolysis at 90 °C for 2 h [10]. β-Galactosidase exhibits specificity and can efficiently hydrolyze glycosidic bonds under mild conditions, which is likely the primary reason why enzymatic methods yield PGPs with lower molecular weights.

### 3.6. Effect of TPGP and EPGP on Emulsion Stability of Whey Protein Isolate (WPI)

The stability of emulsions was evaluated using centrifugation stability and storage temperature stability tests. Centrifugation is commonly employed as an accelerated stability test to trigger dispersion or aging in emulsions rapidly [35]. WPI-GA, WPI-EPGP, and WPI-TPGP demonstrated higher ES values (25.22%, 30.63%, and 33.34%) compared to WPI (11.55%) (Figure 6A). Both TPGP and EPGP at 3% (*w*/*w*) markedly increased the ES of the peach gum-protein complexes (*p* < 0.05). Among these, WPI-TPGP showed the highest ES, which is likely due to the viscosity of the emulsion solution [10]. The viscosity of WPI-TPGP (3% *w*/*w*) is higher than that of WPI-EPGP (3% *w*/*w*), and the viscosity was directly correlated with molecular weight (Table 1). The interaction between WPI and TPGP resulted in a higher viscosity of PGPs in the emulsion, which improved the ES of WPI.

To further explore the impact of PGPs on the long-term stability of emulsions during storage at different temperatures, emulsions were stored at low (4 °C), room (25 °C), and high (45 °C) temperatures for a period of 5 weeks. The storage stability after 5 weeks was ranked as follows: WPI-TPGP > WPI-EPGP > WPI-GA > WPI. Moreover, group WPI-TPGP exhibited better stability across different temperatures. As shown in Figure 6B, emulsions of WPI and WPI-GA stored at different temperatures exhibited varying degrees of rapid phase separation within 24 h. At 4 °C (Figure 6(B-1)), Group WPI exhibited slow phase separation within the first 2 weeks and became stable thereafter. Group WPI-GA showed gradual phase separation during the first week and then remained stable. Group WPI-EPGP remained stable for 5 days, followed by rapid phase separation within the next 3 days, and became stable thereafter. Group WPI-TPGP remained stable for the first week, showed slow phase separation during the second week, and became stable thereafter. At 25 °C (Figure 6(B-2)), Group WPI experienced a sharp decline in stability during the first week but then stabilize from the second week. Group WPI-GA remained stable for the first week, exhibited slow phase separation during the second week, and became stable thereafter. Group WPI-EPGP showed significant phase separation during the first three days, followed by linear phase separation, which resembled the trend observed for Group WPI-TPGP. At 45 °C (Figure 6(B-3)), Group WPI and WPI-GA exhibited rapid phase separation during the first two weeks, followed by stabilization, while Group WPI-EPGP and WPI-TPGP showed rapid phase separation during the first week and became stable thereafter. Emulsions containing gum–protein complexes exhibited higher stability, aligning with centrifuge stability results. These findings demonstrate that PGPs notably (*p* < 0.05) improved the ES of WPI.

Previous studies on tree gum have fully demonstrated that gum polysaccharides exhibit relatively good emulsifying properties, serving as both emulsifiers and stabilizers [36,37,38]. The mechanism primarily involves increasing viscosity and stabilizing oil droplets by adsorbing onto their surfaces, thereby preventing coalescence and ensuring stable dispersion within the emulsion [39]. The prepared emulsions were observed using a BX53 microscope (Olympus, Tokyo, Japan) (Figure 7). Emulsions without GA and PGPs exhibited dispersed oil droplet distribution with irregular morphology. After the addition of GA, the oil droplets became uniform in shape but larger in size. In contrast, the addition of PGPs resulted in oil droplets with more consistent size and distribution without noticeable aggregation. Furthermore, emulsions containing TPGP featured smaller droplet sizes. It has shown that higher stability is generally observed in emulsions with smaller droplets, as smaller droplets both adsorb more polysaccharide molecules on their surface and prevent aggregation and sedimentation effectively [40]. WPI droplets can be coated with a secondary protective layer formed by polysaccharides [41]. Consequently, emulsions containing TPGP and EPGP demonstrated higher viscosity and a more compact fibrillar structure, significantly enhancing the emulsion stability of WPI. The molecular weight of PGPs was positively correlated with the ability to improve oil-water emulsion stability. The viscosity and stability of the emulsion increase with the molecular weight of PGPs, which was consistent with the conclusion of Wei et al. [10]. The functional groups anchored at the oil-water interface and the long molecular chains extending away from the interface are fundamental characteristics of emulsifying stabilizers. PGPs can still function effectively as stabilizers even with low hydrophobic group content, provided the potential anchoring sites are readily accessible and able to bind firmly, facilitating polymer attachment as highly branched polymers [42]. Furthermore, the presence of charged -COOH groups in PGPs enabled electrostatic interactions, which contribute to emulsion stability, particularly under low ionic strength conditions [43]. The highly branched macromolecular structure was likely the main reason for the better emulsifying properties of PGPs, while the spherical morphology of macromolecules effectively enhanced the stability of emulsions at the oil-water interface.

## 4. Conclusions

This study extracted two distinct fractions of peach gum polysaccharides (PGPs) from peach gum using thermal extraction (90 °C) and enzymatic treatment (176 U/g), and the yields of TPGP and EPGP were 52.3% and 46.8%. Monosaccharide composition analysis and FTIR represented that the composition of PGPs was not significantly affected by the extraction method. Both TPGP and EPGP were arabinogalactan (AG) containing hydroxyl, carboxy. TGA revealed that TPGP and EPGP contain two fractions with distinct molecular features. PGP fractions are different in molecular weight and conformation parameters. EPGP, in particular, has a lower molecular weight compared to TPGP. These differences lead to variations in their rheological properties and emulsion stability. PGP solutions generally exhibit shear-thinning flow behavior, with a pronounced shear-thickening effect observed at low shear rates. The apparent viscosity of PGP solutions increases with concentration and decreases with temperature. Moreover, TPGP solutions exhibit higher viscosity than EPGP solutions at the same concentrations (2%, 4%, and 6% (*w*/*w*)), owing to the formation of a stronger network structure, which results in more elastic flow behavior. Compared to gum arabic (GA), emulsions containing PGPs exhibited better stability, as PGPs adsorb onto oil droplets, stabilizing the particles or droplets dispersed in the emulsion and preventing flocculation and coalescence.

## Figures and Tables

**Figure 1 foods-14-03341-f001:**
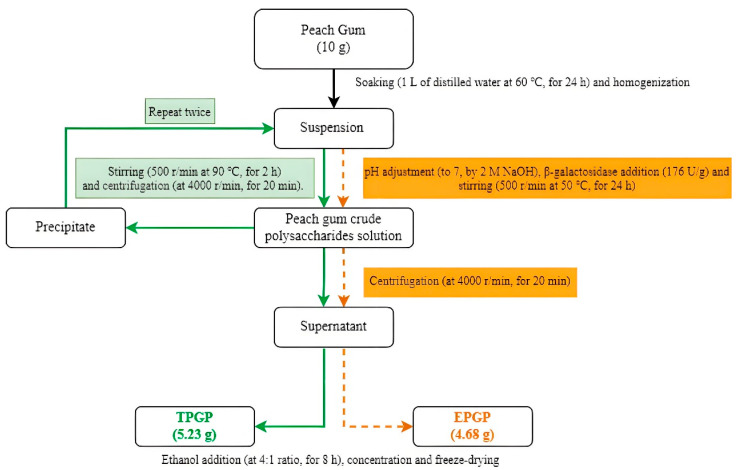
Extraction and purification of TPGP and EPGP.

**Figure 2 foods-14-03341-f002:**
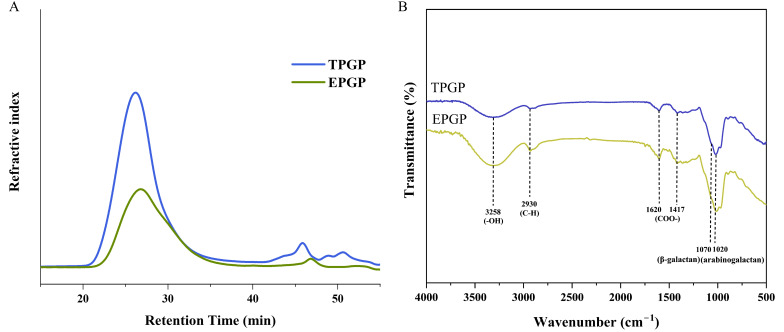
HPSEC profile (**A**) and FT-IR spectroscopy (**B**) of TPGP and EPGP.

**Figure 3 foods-14-03341-f003:**
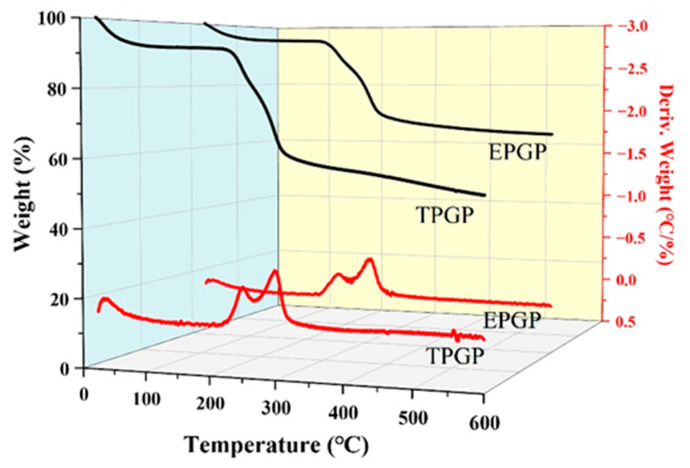
Thermogravimetric Analysis (TGA)/Derivative Thermogravimetric Analysis (DTGA) curves of TPGP and EPGP.

**Figure 4 foods-14-03341-f004:**
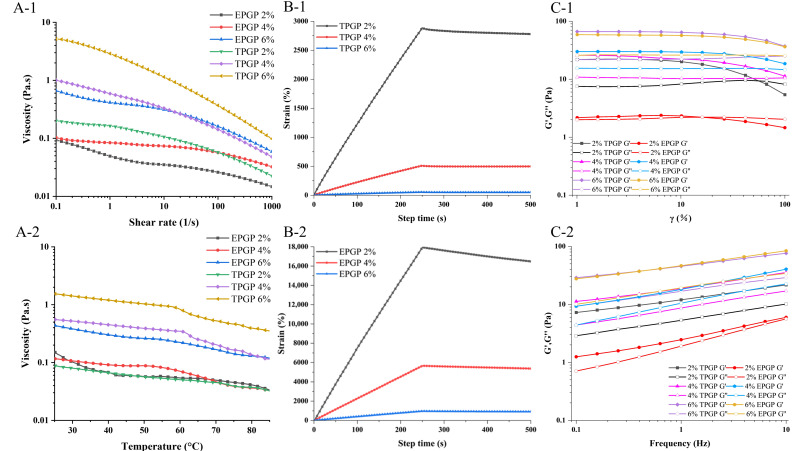
Steady shear tests of PGP solutions: Shear rate range of 0.1 to 1000 s^−1^ at 25 °C at different concentrations and treatments (**A-1**), under the heating process, the temperature was increased from 25 °C to 85 °C at a rate of 5 °C/min, with a shear rate of 10 s^−1^ (**A-2**); The effect of TPGP and EPGP concentrations on the creep and recovery behavior at 25 °C (**B-1**,**B-2**); Dynamic rheological tests: the effects of PGPs under different concentrations and treatments on G′ and G′′ as a function of strain, 1–100%, 25 °C, 1 Hz (**C-1**); the effects of PGPs concentrations on G′ and G′′ as a function of frequency, 0.1–10 Hz, 25 °C, the strain of 10% (**C-2**).

**Figure 5 foods-14-03341-f005:**
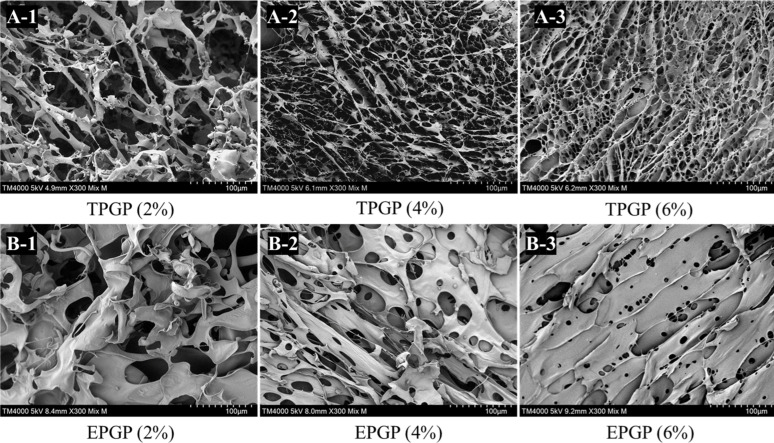
The microstructure of TPGP (**A-1**–**A-3**) and EPGP (**B-1**–**B-3**) gel with different concentration under different concentrations by SEM.

**Figure 6 foods-14-03341-f006:**
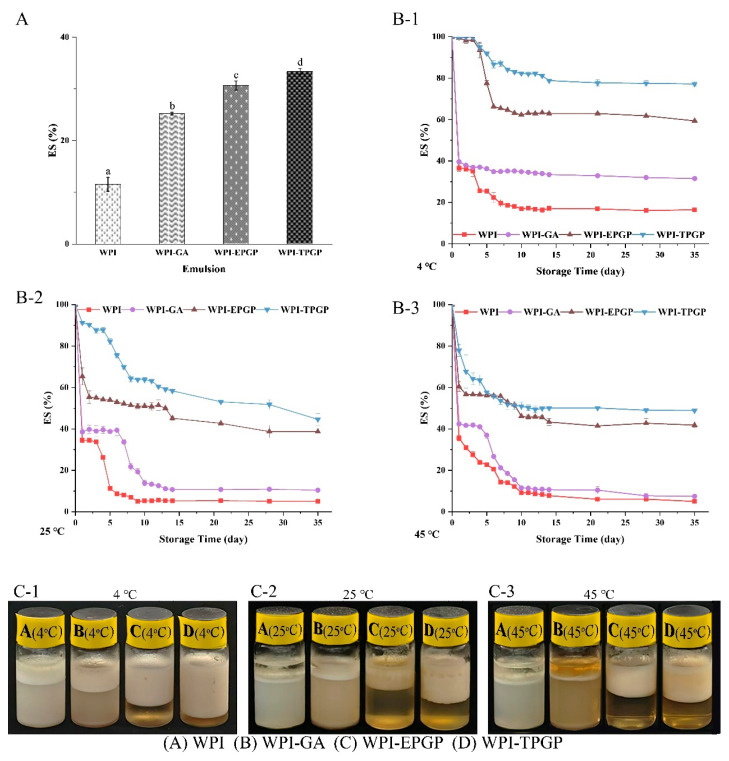
The effects of peach gums on the stability of WPI stabilized oil–water emulsions: Centrifuge stability of WPI emulsions stabilized with PGPs (**A**) lowercased letters denote statistically significant differences *p* < 0.05; Storage stability of WPI emulsions stabilized with PGPs at 4 °C (**B-1**), 25 °C (**B-2**), 45 °C (**B-3**) in 5 weeks; Appearance of WPI emulsions stabilized with PGPs and GA after storage for 5 weeks at 4 °C (**C-1**), 25 °C (**C-2**), 45 °C (**C-3**).

**Figure 7 foods-14-03341-f007:**
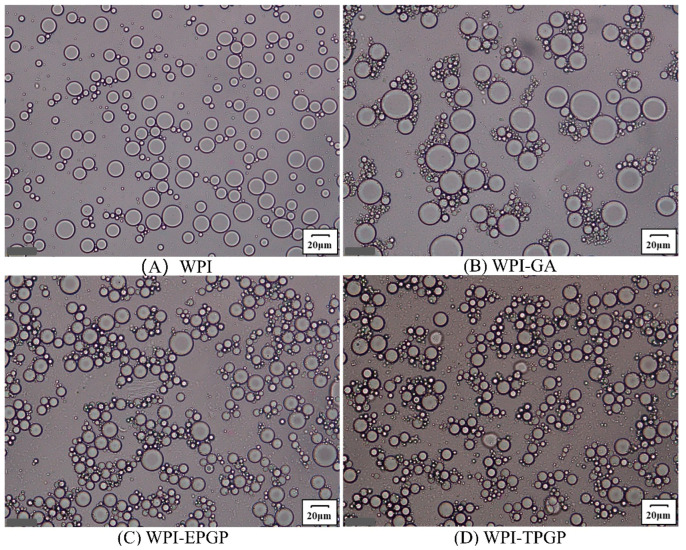
Microscope structure of droplets in WPI emulsions stabilized (scale bar = 20 µm, magnification 26.2×): (**A**) WPI, (**B**) WPI-GA, (**C**) WPI-EPGP, (**D**) WPI-TPGP.

**Table 1 foods-14-03341-t001:** The chemical properties of PGPs.

Samples	TPGP	EPGP
Yield % (*w*/*w*)	52.3 ± 2.10	46.8 ± 1.64
Total sugars % (*w*/*w*)	70.62 ± 1.38	72.13 ± 2.07
Uronic acid % (*w*/*w*)	4.34 ± 0.09	2.13 ± 0.06
Monosaccharide compositions % (molar ratio)		
Rha	0.78	0.69
Ara	51.20	50.64
Gal	29.93	30.96
Xyl	10.27	9.68
Man	4.59	4.49
Glc-UA	3.23	3.53

**Table 2 foods-14-03341-t002:** Molecular weight and conformational parameters of PGPs.

Samples	TPGP	EPGP
*M*_n_ (g mol^−1^)	6.491 ± 0.03 × 10^6^	2.763 ± 0.02 × 10^6^
*M*_w_ (g mol^−1^)	1.295 ± 0.05 × 10^7^	5.718 ± 0.02 × 10^6^
*M*_w_/*M*_n_	1.994 ± 0.06	2.070 ± 0.03
*R*z (nm)	240.7 ± 0.02	147.9 ± 0.01

## Data Availability

The original contributions presented in the study are included in the article; further inquiries can be directed to the corresponding author.

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
