# Peer review of "Rheological Properties and Emulsion Stability of Peach Gum Polysaccharides with Different Molecular Weights"

_foods, 2025, doi:10.3390/foods14193341_

Round 1

Reviewer 1 Report

Comments and Suggestions for Authors

Reviewer’s Report

This manuscript investigates the rheological behavior and emulsion-stabilizing properties of peach gum polysaccharides (PGPs) with different molecular weights, comparing thermally and enzymatically extracted samples. The study is relevant to food hydrocolloid applications and demonstrates strengths such as a clear comparative design, diverse analytical techniques (HPSEC–MALLS, FTIR, rheology, TGA), and potential industrial relevance in stabilizing whey protein isolate emulsions. However, several inconsistencies, methodological gaps, and presentation issues limit the clarity and reliability of the current version. I recommend minor revision rather than rejection, as the work is fundamentally sound and requires minor corrections and clarifications to meet publication standards. 

  • Reference #7 (lines 501–502) should be corrected. At the moment, it cites “Yang et al., 2018 International Conference on Applied Engineering,” which is an electrical engineering paper and unrelated to PGP hydrolysis at 140 °C. Please replace this with the correct peach gum hydrolysis source that reports a molar mass of 6.8 × 10⁵ g mol⁻¹ under those conditions.
  • Reference #1 (lines 489–490) needs clarification. The title currently reads “peach gum from almond (Prunus dulcis).” If the material used in that study is almond gum and not peach gum, it should be presented as a different system, with clear caveats before using it as precedent.
  • The order in which tables and figures are cited should be checked. As noted, Table 2 (lines 257–260) is not cited in numerical sequence. Please make sure all tables and figures are first mentioned in the correct order in the text.
  • On line 1, the document label should be completed so it reads “Type of the Paper (Article)” with the closing parenthesis in place.
  • The title (lines 1–3) should be revised to “Rheological Properties and Emulsion Stability …” to avoid the comma splice currently present in “Rheological Properties, Emulsifying Stability ….”
  • In the author block (lines 4–17), please standardize the punctuation and spacing in emails and initials—for example, “sihaoyu1023@126.com (H.S.)” and “sundapeng@saas.sh.cn (D.S.)”—and correct any mis-ordered affiliation superscripts such as “Zhan Lin 6, 4.”
  • In the Abstract (lines 20–36), maintain a consistent present tense, changing “The results indicated that … exhibit” to “The results indicate that … exhibit.” Also, change “As PGPs concentration increased” to “As PGP concentration increased,” use proper SI formatting “1.295 × 10⁷ g mol⁻¹” instead of “1.295×10⁷ g/mol,” and replace “Emulsifying stability” with “Emulsion stability,” making the same change in the keywords.
  • In the Introduction (lines 41–49), use the present tense—“Peach gum is rich …” rather than “was rich ….”
  • Still in the Introduction (lines 58–66), replace reference #7 with the correct hydrolysis source and confirm that the conditions (140 °C for 2 h) and molar mass (6.8 × 10⁵ g mol⁻¹) truly apply to peach gum.
  • In the Methods (lines 111–122), specify the exact ethanol-to-supernatant ratio (v/v) and describe how the pellet was handled between extractions. Also, change “And The extraction rate …” to “The extraction yield ….”
  • In the monosaccharide analysis section (lines 126–139), list the standards used, describe the calibration method, and explain the chromatogram integration criteria.
  • In the HPSEC section (lines 141–148), provide the dn/dc value, the column set, the MALLS angles, the analysis software used, and the temperature. Clarify whether proteins or phenolics were removed before analysis.
  • In the rheology methods (lines 162–170), give details of the cone or plate geometry (angle, truncation, or gap), state whether a solvent trap was used to prevent evaporation, and note the pre-shear and equilibration times. The steady-shear range given here should match the one shown in Figure 4.
  • In the emulsion preparation (lines 187–195), state the pH, ionic strength, and buffer composition of the aqueous phase; give the exact concentration of pentanediol; and report either the homogenization energy (kJ kg⁻¹) or a full description of rotor speed, time, and rotor diameter. Also mention the emulsion temperature after homogenization.
  • In the stability section (lines 196–205), define Ve clearly (is it the cream layer or the total emulsion height?), give tube dimensions and describe the reading protocol, and increase the number of replicates to at least three.
  • In the Results (lines 228–236), replace the contradictory phrase “highly homogenous … and broad distribution” with “moderately polydisperse” to reflect Mw/Mn ≈ 2.
  • In the Results (lines 237–244), remove the explanation that attributes differences to electrostatic repulsion and replace it with a mechanism that fits the screened conditions of your experiment.
  • In the FTIR discussion (lines 248–257), avoid over-interpreting peaks such as assigning “furanose rings” to the 773/710 cm⁻¹ bands unless supported by composition data and suitable references.
  • In the TGA section (lines 265–272), clarify whether both samples display two distinct mass-loss events, and report the DTG peak temperatures and the mass loss for each fraction separately for TPGP and EPGP.
  • In the flow behavior discussion (lines 289–309, around line 305), remove the general Newtonian claim or restrict it to very dilute regimes, providing supporting data.
  • In the oscillatory rheology section (lines 334–354), report the LVR limits for each system and justify the use of 10% strain for frequency sweeps by showing it lies within the LVR for all samples.
  • In the discussion of thermal degradation (lines 362–371), change “high temperatures inducing proton formation” to “autohydrolysis/thermal degradation” and add an appropriate citation, or remove the statement.
  • In the emulsion results (lines 421–447), support droplet-size and structure claims with quantitative analyses such as laser diffraction, and include representative micrographs with scale bars and full size distributions.
  • In the Conclusions (lines 458–466), remove the statement about a “pronounced shear-thickening at low shear rates” and ensure all conclusions are consistent with the corrected rheology findings.
  • In the Funding section (lines 471–476), correct “great number” to “grant number” and avoid repeating funding details in the Acknowledgments.
  • In the Data Availability statement (line 477), correct the punctuation so it reads “included in the article; further inquiries ….”
  • In the Acknowledgments (lines 479–484), remove redundant content and follow the journal’s style.
  • On line 122, fix the capitalization so it reads “The extraction yield ….”
  • In lines 393–405, replace “verged to stabilize” with “then stabilized” and standardize the way temperatures are written, for example “25 °C.”
  • Throughout the manuscript, ensure proper SI spacing, use consistent units (g mol⁻¹, w/w), and avoid subjective terms like “excellent stability” unless these are backed by statistical evidence.

Reviewer 2 Report

Comments and Suggestions for Authors

see report

Round 2

Reviewer 2 Report

Comments and Suggestions for Authors

The manuscript has been sufficiently improved to warrant publication.

Author Response

Thanks for your suggestion.